# Determining Critical Thresholds of Environmental Flow Restoration Based on Planktonic Index of Biotic Integrity (P−IBI): A Case Study in the Typical Tributaries of Poyang Lake

**DOI:** 10.3390/ijerph20010169

**Published:** 2022-12-22

**Authors:** Zhuowei Wang, Wei Huang, Dayu Zhu, Qi Huang, Leixiang Wu, Xingchen Liu

**Affiliations:** 1State Key Laboratory of Simulation and Regulation of Water Cycle in River Basin, China Institute of Water Resources and Hydropower Research, Beijing 100038, China; 2Department of Water Ecology and Environment, China Institute of Water Resources and Hydropower Research, Beijing 100038, China; 3Key Laboratory of Poyang Lake Wetland and Watershed Research, Ministry of Education, Nanchang 330022, China; 4School of Geography and Environment, Jiangxi Normal University, Nanchang 330022, China

**Keywords:** Poyang Lake, environmental flow, hydrological mutation, Fuhe River Basin, planktonic index of biotic integrity

## Abstract

Hydropower construction and climate change have aggravated river hydrological changes, which have reduced the water flow regime in the Ruhe River Basin. The reduced flow of the river seriously affected the water supply of nearby residents and the operation of the river ecosystem. Therefore, in order to alleviate the contradiction between water use for hydropower facilities and environmental water use, the urgent need is to explore the ecological flow-threshold of rivers. This study took the Fuhe River Basin as the research object, and summarized the monitoring data of eight hydrological stations from recent decades. Based on this, we explored the response law of P−IBI and flow, a tool to quickly measure the health of the ecosystem. Through the response relationship between alterations in environmental factors of the river and phytoplankton index of biotic integrity (P−IBI), it was determined that environmental flow was the dominant influencing factor of P−IBI. According to P−IBI, the threshold of environmental discharge in the Fuhe River was limited to 273~826.8 m^3^/s. This study established a regulatory framework for the river flow of large rivers by constructing P−IBI and determining the critical thresholds of environmental flow by constraining the constitution. These results provide a theoretical basis for better planning and improvement of river ecosystem restoration and river utilization.

## 1. Introduction

The Fuhe River, one of the main rivers in the Poyang Lake water system, plays an important role in sustaining the public water supply and aquatic ecosystem [1]. Unfortunately, various human activities and global climate change have reduced seriously the flow regime in the Fuhe River. As a consequence, there have even been several persistent large-scale dry-water events. Specifically, the sharp reduction of environmental flow has attributed to the construction of hydropower in the Fuhe River Basin. Therefore, the environmental flow is plagued by flow reduction in the Fuhe River [2,3], which has further exacerbated the severity of the water-level drop in Poyang Lake. Furthermore, from July to October, when rainfall decreases and evaporation increases, the surrounding farmland needs so much water irrigation that it squeezes the ecological water of the river. This happens to be the time when the Fuhe River Basin has the least water flow. This leads to the contradiction between ecological water use and public water supply. Inadequate management of water projects may affect the water use of residents and the normal operation of ecosystems, such as the destruction of animal habitats, water-quality degradation, and soil erosion [4,5,6]. Therefore, determining the threshold value for the requirement of the environmental flow is an urgent problem to be solved at present.

Environmental flow refers to the flow magnitude, frequency, timing, duration, and velocity of alteration that are retained in rivers to maintain the operation of river ecosystems. After realizing the existence and impact of these threats, humanity has conducted extensive research to restore rivers [7,8]. The research results show that the community diversity of river microorganisms, plants, and animals can be effectively enhanced by regulating the environmental flow of the river [9,10,11,12]. Maintaining the environmental flow of the river will help to continuously improve the water-environment quality of the river and help to restore the health of the river and the surrounding ecosystems [13,14]. However, constructing the threshold of environmental flow has always been a difficult problem, because of the variable requirements of the components of the ecosystem. Therefore, constructing an evaluation system is a vital process which can comprehensively evaluate the operation and health of a large river basin.

The index of biotic integrity (IBI) is widely regarded as a tool to evaluate the health of river ecosystems. Karr [15] first devised the IBI to measure biological integrity in streams, mainly using fish as an indicator species. Moreover, the Karr index has been adopted by many management agencies and is even used to assess the integrity of estuarine ecosystems [16,17,18]. However, it is difficult to use fish as indicator species when the flow of a river is reduced or even stopped. All components of the water ecosystem function are primarily affected by phytoplankton and zooplankton [19,20,21]. Phytoplankton are the main source of energy driving large aquatic ecosystems such as mussels and clams [22,23], and they also respond quickly to environmental change. The downstream decline in phytoplankton is attributed to low water flow, which also leads to impaired water quality [24]. Kim et al. [25] found that there was a certain dynamic mechanism of phytoplankton changes with the water flow. Therefore, this study developed the plankton integrity index (P−IBI) to indicate the health of the river ecosystem. Therefore, this paper developed the relationship between the P−IBI and the flow to obtain the flow threshold that can maintain river health. The study aims are (1) to establish a quantitative relationship between the P−IBI and the flow regime of Fuhe River, (2) to explore the range of environmental flow thresholds, and (3) to meet the environmental-flow-guarantee requirements of the Fuhe River.

## 2. Materials and Methods

### 2.1. Study Region and Sampling Sites

The Fuhe River Basin (115°35′~117°09′ E, 26°30′~28°50′ N) is located in the southeast of Jiangxi Province and is one of the five major tributaries of the Poyang Lake, the largest freshwater lake. It originates from Guangchang County, Fuzhou City, Jiangxi Province, at the western foot of the Wuyi Mountains. It is the second largest river in Jiangxi Province, with a total length of 349 km and a drainage area of 171.86 million square kilometers. Parameters such as phytoplankton biomass, species number, the class of phytoplankton, flow regime, etc. were used to develop the quantitative relationship between the P−IBI and flow regime. The locations of sampling points and hydrological stations are shown in Figure 1.

### 2.2. Hydrological Alteration in Fuhe River Basin

To identify the year when the hydrological mutation abruptly occurred, and to analyze the hydrological alteration degree before and after the mutation, the MK test [26,27,28] and the sliding *t*-test were used in this study to analyze the Shaziling (SZL), Liaojiawan (LIA), and Lijiadu (LID) parts of the mainstem of the Fuhe River. The hydrological mutation of the flow at the hydrological stations of Shuangtian (ST), Xinxie (XX), and Gongxi (GX) the main part of the hydrological station and tributaries, as well as the Taopi (TP) and Loujiacun (LOU) hydrological stations in the downstream, were identified over the year.

To evaluate the hydrological alteration, this study is based on the RVA method proposed by Richard et al. [26,29,30]. The RVA method divides the discharge time-series into two parts, before and after hydrological mutation. According to the 33 IHA indicators calculated from the flow time-series before the hydrological variation, the 33 indicators are divided into three intervals, in accordance with the ranking percentage of each indicator: L1 = [0, 25%], L2 = [25%, 75%], L3 = [75%, 100%], where the target interval is L2, and the alteration IAi of each IHA index after mutation is calculated. The calculation formula of IAi is:IA_i_ = |N_i_ − N_ε_|/N_ε_ × 100%
$$\mathrm{IA}=1/33\overset{33}{\underset{i=0}{\sum }}{\mathrm{IA}}_{i}$$

Among these, i is the serial number of the index, N_ε_ the number of years falling into the target range before the impact, N_i_ is the number of years falling into the target range after the impact, which is 50%, and IA is the overall degree of hydrological alteration. When IA is in the range of [0, 0.33], it is a low disturbance state, [0.33, 0.67] is a medium disturbance state, and [0.67, 1] is a high disturbance state [27,28,31].

When the calculated and analyzed hydrological-data-event sequence is more than 20 years, climate change can reduce or eliminate the impact on the results of hydrological indicator calculations [32,33,34]. Therefore, the time series of the hydrological data calculated and analyzed using the IHA method should generally not be less than 20 years (Appendix A).

The geographic location of the hydrological station is shown in Figure 1. The IHA and RVA methods were used to calculate the hydrological mutation of the hydrological stations in the Fuhe River Basin before and after the hydrological mutation.

### 2.3. Metric Selection

For constructing the P−IBI evaluation system, twenty-six phytoplankton indicators of the types of population, nutrient-structure, algae bloom characteristics, diversity, and evenness were used as candidate indicators (Table 1). All these data were obtained from the field survey and used in laboratory quantitative analysis. After the candidate indicators were counted, using distribution range analysis, discrimination ability analysis, normal distribution test, correlation analysis, etc., the indicators with too small or too large a distribution range, insignificant significance, poor discriminative ability, and highly overlapping information, were eliminated. Finally, the indicators that passed the screening were used as the core indicators.

The distribution-range screening was performed by eliminating the parameter in which the value of 95% of the sample points was 0. The discrimination ability analysis was determined by the discrepancy between reference points and damaged points, referring to the method of Barbour et al. [35] The Pearson correlation analysis was performed on the selected indicators. For indicators with significant correlation, one indicator in a group was selected and left. Finally, 14 indicators, including M1, M2, M9, M10, M11, M12, M19, M20, M21, M22, M23, M24, and M26 were left to construct the P−IBI evaluation-index system.

The P−IBI values were calculated using the ratio method: first, calculate the value of the 95% or 5% quantile of each core index for all sample points. For the index whose value decreases with the increase of interference, the index value of the 95% quantile is the best-expected value. Indicator score = measured value/best expected value; for indicators whose value increases with the increase of interference, the indicator value of the 5% quantile is the best expected value, and indicator score = (maximum value − measured value)/(maximum value − best expected value). Finally, the core index-scores are added to obtain the total P−IBI score for each sample point.

### 2.4. Phytoplankton Collection and Analysis

The phytoplankton was collected with a no. 25 plankton collection net. Before sampling, the piston was opened to clean the plankton net, and 5 L water samples were collected with a quantitative water collector and poured into the plankton net for filtration and concentration. Two drops of Lugol’s solution were added to the collected samples, and thoroughly mixed. The water needed to be added into the samples up to 50 mL before the laboratory analyses, and 0.1 mL of this solution was put into a 0.1 mL counter chamber. The whole chamber was counted under a 10× 40× microscope, and each sample was counted 2–3 times. Assuming a specific gravity of 1.0, the phytoplankton abundance was converted to biomass. The biomass was calculated by multiplying the phytoplankton abundance by the average wet weight of the corresponding volume.

### 2.5. Statistical Analysis

The IHA of each hydrological station in the Fuhe River Basin was calculated using IHA software (The nature conservancy, London, UK). The IHA information and the alteration components in the hydrological years of 8 hydrological stations were summarized. Based on the measured hydrological data of the watershed and the physical and chemical properties of the river, the Spearman correlation coefficients between the phytoplankton biomass and flow, temperature, nutrients, electrical conductivity, pH, etc. were analyzed, mainly based on SPSS 19.0 (SPSS, Chicago, IL, USA). Origin 2021 was used to fit and evaluate the fitting of P−IBI and environmental flow. Origin 2021 was used for regression analysis, and the confidence interval of 95% confidence was calculated.

## 3. Results and Discussion

### 3.1. Hydrological Alteration Identification in Fuhe River Basin

According to the mutation test analysis, due to the completion of the Qingtong Reservoir and the Chemoling Reservoir, the mutation points of the SZL and ST located in the upper reaches of Fuhe River appeared in 1987 and 1979, respectively. Due to the alteration of water consumption in TP irrigation area, TP and XX in the middle reaches of Fuhe River both mutated in 1991. The hydrological mutation of GX was in 1979, which was mainly related to the construction and operation of the power stations in the middle reaches of the Fuhe River. LOU, LID, and LIA are located near the entrance of Poyang Lake in the lower reaches of the Fuhe River. The mutation time may be significantly affected by the construction of the hydropower project in the lower reaches of the Fuhe River in the 1970s and 1980s. The construction of dams has resulted in reduced runoff and reduced water flow into rivers, causing severe damage to aquatic ecosystems and the surrounding soil ecosystems [36,37,38]. In addition, the Fuhe River Basin often has a great demand for irrigation with massive water withdrawal, which has various impacts on the surrounding environment, such as habitat destruction, water-quality degradation, soil erosion, etc. [39,40,41].

### 3.2. Hydrological Alteration Assessment in Fuhe River Basin

Based on the divided hydrological-sequence-change nodes, the IHA software was used to calculate the IHA degree of each hydrological station in the Fuhe River Basin (Table 2 and Table 3). An analysis of the change characteristics of 33 indicators in 5 groups shows that the alterations of SZL, ST, and downstream LOU, LIA, and LID located in the upstream source section are mainly low-level disturbances, with an alteration degree of less than 33%. There are 22 indicators with low alterations in SZL, 22 in Loujia Village, 18 in LIA, and 20 in Lijiadu. This presents the characteristics of low-level disturbance, which is mainly the result of the combined effect of the presence of tributaries in the upstream and downstream reaches and the coordinated regulation of gates and dams. [42,43,44]. ST in the upper reaches and XX, GX, and TP in the middle reaches of the river are dominated by medium-to-high disturbances. There are 18 indicators with moderate alterations in ST, 20 in XX, 16 in GX, and 15 in TP. In comparison, the hydrological variability of the tributaries is higher than that of the mainstem. The hydrological variation of tributaries is mainly reflected in the monthly average flow and extreme flow from March to June, which shows a significant increase in variation characteristics. The mainstem in the upper and middle reaches of SZL and LIA does not show high variability indicators, while the LID has a significant decrease in the indicators of minimum flow at different time scales, which may affect the overwintering conditions of species in the marsh or reduce the ability of organisms to cope with extreme weather [45,46,47]. In addition, the reduced-flow river can easily cause algal blooms [48,49], which exacerbates the risk of algal blooms in the Fu River Basin. In addition, LID is the last hydrological station that flows into Poyang Lake. Its hydrological indicators and mutations are related to the stable operation of the Poyang Lake ecosystem. Therefore, LID was chosen as the threshold evaluation of the P−IBI environmental flow.

The monthly average flow is shown in Appendix A, and the difference in Figure 2. In the monthly average of the SZL, LOU, LID, and LIA on the mainstem of the Fuhe River from June to September and December, the flow showed a decrease; the decrease in June was the most obvious, and the decrease in September was the weakest. In the 12 months, the SZL and the LIA all experienced moderate-to-low alterations, among which the monthly average-flow alterations of the SZL in May and September were a minimum of 4%. The monthly average-flow alteration of LIA in February was at least 2%. The hydrological alteration of the monthly average flow at LOU in April showed a high alteration of 73%. The hydrological alteration of the monthly average flow of LID in September showed a high alteration of 88%. The rest of the months showed low to moderate alterations. LOU, LID and LIA are located at the exit of the lower reaches of the Fuhe River. There are many hydropower projects which impact the hydrological alteration of the monthly average flow. The monthly average flow of ST, XX, and TP on the tributaries of the Fuhe River decreased from June to December, and the monthly average flow of GX from June to September, November, and December showed a decrease. In particular, the decrease in June was the most obvious, and the decrease in November was the weakest. In the 12 months, ST, XX, GX, and TP accounted for half of the low-to-medium alteration and high alteration; the high alteration of ST in April and June was 83%, and the minimum alteration in January was 8%. The hydrological alteration of XX in March, May, and June was 72% in height, and the hydrological alteration in September and December was at least 3%. The high alteration of TP in March and May was 128% and 68%, respectively. The monthly average-flow alteration in January and October was the lowest, at 8%. The monthly average flow of GX in April and October showed a high degree of alteration. The largest alteration was the monthly average flow in October, which reached 92%. The monthly average-flow alteration in February was at least 4%. In short, since the SZL is located at the confluence point of the tributaries in the upper reaches of the watershed and downstream of the LIA, the monthly average-flow alterations of the two were mainly low to moderate. However, the height variation of other hydrological stations is attributed to the regulation and operation of hydropower projects and dam building in the Fuhe River Basin [50].

After the disturbance, the hydrological alterations of the annual extreme flow at the SZL, LOU, and LIA on the mainstem were all low-to-moderate alterations. The hydrological alterations of the 1-day minimum flow and the 3-day maximum flow were at least 4%. The hydrological alteration of the 90-day maximum flow was the highest, reaching 46%. The hydrological alteration of the 1-day minimum flow at LOU was the lowest, at 2%, and the hydrological alteration of the 30-day minimum flow was the largest, reaching 61%. LIA had the lowest alteration in 30-day maximum flow, which was also 2%. The hydrological alteration of the 7-day and 30-day minimum flow was the largest, which was the same as the hydrological alteration of the 30-day minimum flow at LOU, which was 61%. This shows that the construction of hydropower projects has the same impact on the extreme value of the daily flow at LOU and LIA. The hydrological alterations of the extreme flow at the ST on the tributaries are all low-to-moderate, and the maximum alteration of the daily maximum flow is the largest, reaching 58%. The hydrological alterations of the 7-day minimum flow, 7-day maximum flow, and 90-day maximum flow were the lowest, and were both 8%. Among the hydrological alterations of the extreme flow at XX, GX, and TP, the mid-altitude alteration accounted for half of the hydrological alterations, and the 90-day maximum flow at XX and TP both showed high alterations. The hydrological variation of the TP daily minimum discharge was a maximum of 100% hydrological variation of the extreme discharge in the tributaries.

After perturbation, the upstream SZL extreme-flow date has a tendency to delay in the upstream SZL, contrary to ST. The date of the extreme flow of each hydrological station in the midstream shows a tendency to be delayed. The dates of extreme flow at the downstream LID and LIA tend to be delayed, but the extreme flow period of LOU tends to be advanced.

From the pulse times of high and low flow and the hydrological alteration over time, it can be seen that the hydrological alterations of extreme flow at SZL, ST, LOU, and LIA are all low-to-moderate alterations after disturbance. The number and duration of high- and low-flow pulses at LID, XX, GX, and TP have altered in height, but only once or twice, such as twice in TP. The occurrence time of the maximum traffic of TP and GX varies greatly, especially where TP reaches 104%. The hydrological alterations of the average flow rise/fall rate, and the reversal times of the hydrological stations on the mainstem after disturbance were mainly low to moderate. The average flow fall rate of the SZL had a high alteration degree of 68%, and the number of reversals also increased slightly. The hydrological alteration degree of the reversal times of the ST, XX, and TP on the tributaries all showed a high degree of alteration, reaching more than 70%.

We summarized the IHA information and the altered components of the hydrological indicators per year for eight stations, and the results of the overall hydrological alterations of each section are shown in Figure 3. XX, TP, GX, LOU, ST, LID, and LIA are all in a moderate disturbance state. From upstream to midstream, and then from midstream to downstream, the hydrological disturbance is weak, from strong. The fluctuation range from upstream to midstream is 0.31~0.55, and the fluctuation range from midstream to downstream is 0.36~0.55. The TP and XX of the tributaries have the strongest disturbance, and the SZL upstream of the mainstem has the weakest disturbance. The mutation rate and degree of the mainstem hydrological regime in the Fuhe River Basin were relatively low. However, the increased demand for domestic water from hydropower projects and surrounding residents has resulted in a decrease in tributary flows [51,52]. River populations (such as phytoplankton and zooplankton) respond to alterations in flow [53,54]. In recent years, the construction of dams in the Fuhe River Basin and extreme drought conditions have occurred from time to time. Inappropriate flow regulation in the lower reaches of the river basin will cause the ecosystem to operate in a disorderly way. Therefore, it is a requisite to maintain the flow alteration in an appropriate range, called the environmental flow, to preserve the health of the ecosystem of the river. This study develops an approach to evaluate the environmental-flow-threshold range based on the response of the P−IBI to the flow of the Fuhe River.

### 3.3. Spatial and Temporal Distribution of Phytoplankton

A total of 128 species of phytoplankton were detected in Fuhe River Basin, including 51 phyla of *Bacillariophyta*, accounting for 39.8% of the total phyla, 51 species of *Chlorophyta*, accounting for 39.8% of the total phyla, and 15 species of *Cyanobacteria*, accounting for 11.7% of the total phyla. Among them, *Anabaena pseudobaena* and *Anabaena granulosa* are the dominant phyla in the Fuhe River Basin.

The phyla and quantity of phytoplankton in Fuhe River show significant differences in different-flow river reaches from upstream to downstream; the number of phytoplankton species is upward, the number of *Chlorophyta* is gradually increasing, and the number of *Bacillariophyta* is gradually decreasing (Figure 4). In summer (July), with the appropriate water temperature and sufficient light, the phytoplankton species in the Fuhe River Basin is at a peak. From upstream to downstream, the dominant species is not significantly different in the wet season, while in the dry season and in the normal flow season, the phytoplankton species decreased significantly at each sampling point. The biomass of phytoplankton in the wet season is much higher than in the other two periods. The proportion of biomass of the dominant phytoplankton species (*cyanobacteria* and *chlorophyta*) is also much higher than that in normal and reduced-flow river periods. The sensitivity of phytoplankton to river flow and the feasibility of constructing P−IBI were verified.

### 3.4. Response of Phytoplankton to Environmental Indicators

Phytoplankton are affected by multiple factors, such as temperature, light, nutrients, flow rate, etc. To evaluate the impact of various major environmental factors on phytoplankton, based on the measured hydrological data and the basic indicators of the river, the Spearman correlation coefficients between phytoplankton biomass and flow, temperature, nutrients, electrical conductivity, pH, etc. were analyzed. Since biological growth has a cycle, there may be a certain hysteresis in response to environmental alterations, in this study; in addition to paying attention to the current flow during the ecological survey, the average flow of the previous month and the previous two months was also considered. There was a significant linear correlation between phytoplankton and the 1-month, 2-month, and 3-month average flow (*p* < 0.05), and the correlation with the 2-month average flow was the most significant (*p* < 0.01); however, there was no significant correlation between other environmental factors and phytoplankton biomass (Figure 5). The results showed that flow was the main factor affecting phytoplankton biomass. With the increase in flow, phytoplankton biomass showed a positive ecological response, showing an overall upward trend [55,56,57,58]. In the early stage of river restoration, flow restoration will promote the growth and metabolism of primary producers, thereby promoting the ecological restoration of the entire ecosystem [59,60].

### 3.5. Quantitative Response Relationship between P−IBI Index and Flow

In accordance with to correlation analysis in 3.2, the sites with hydrological monitoring data points were selected, and the response relationship between the P−IBI score and the 2-month average flow was constructed. Since the catchment area of each hydrological station in the basin was different, the flow itself had a certain difference. To eliminate the flow difference caused by the catchment area of the river basin, the method of flow/catchment area was used in this study to construct the P−IBI index and the unit set. The quantitative response relationship between the water area and the river flow regime is shown in Figure 6a. The response relationship between the P−IBI score and flow rate is y = (5.94 ± 0.33) × x + (−3743.35 ± 335.40) × x^2^ (R^2^ = 0.84).

This response relationship shows that the health of the watershed ecosystem increases with the increase of flow, and when it reaches a certain peak it decreases with the increase of flow. This is consistent with our prediction. When the flow rate is too low, the hydrodynamic conditions are insufficient, and the minerals and nutrients in the water body accumulate easily, which easily causes algae outbreaks, which in turn affects the health of the river ecosystem; when the flow rate is too high, minerals and nutrients in the water body are easily washed away, phytoplankton lacks the substances needed for growth, and the flow rate is too fast; algae are also easily washed away with the water flow, so it is not conducive to the growth of algae, causing the food of benthic animals, zooplankton and fish sources to be reduced, thereby affecting the health of river ecosystems.

The 95th percentile of all P−IBI scores in the watershed was chosen as the criterion for watershed health to assess the range of flow thresholds when the P−IBI score was at a healthy level (Figure 6b). The value of the 95% quantile of the watershed is 9.896. Solving the one-dimensional quadratic equation of the correlation, the flow range per unit catchment area is 0.0175–0.0530 m^3^/s. For the catchment area of LID, the flow-threshold range is 273–827 m^3^/s. Many studies have shown that damming and hydropower projects in the upper reaches of the lake will lead to a substantial reduction (more than 80%) or even drying up in some river sections, and flow-reduced river reaches downstream will be formed [61,62]. As a result, the downstream hydrological conditions have undergone significant alterations, resulting in serious degradation of the structure and function of the river ecosystem [63,64]. The reduction of flow has a significant negative impact on the aquatic ecosystem and the microbial, plant, and animal-community structures in the flow-reduced reaches. Through the regulation and restoration of downstream flow, the structure and function of the biological community in the downstream ecosystem can be significantly restored [65]. Therefore, in this study, the environmental flow-threshold of the lower reaches of the Fuhe River Basin was set as 312–908 m^3^/s through the ecological response of the LID P−IBI index to the flow. When the dry and wet seasons in the basin alternate, dams and hydropower projects have a feasible flow-threshold range for downstream environmental-flow-regulation.

## 4. Conclusions

Based on the data from eight hydrological monitoring stations in the Fuhe River Basin, this paper summarizes the hydrological situation of the Fuhe River Basin under the influence of human and climatic conditions in recent decades. LID is a hydrological station entering Poyang Lake, and its environmental flow regulation is related to the health and operation of the Fuhe River Basin and Poyang Lake. Therefore, in our study, through constructing the relationship between the phytoplankton index of biotic integrity and the watershed environmental factors, we discovered a rapid tool for evaluating the biological water quality of the ecosystem. The results showed that river flow was the main influencing factor for phytoplankton. In the early stage of ecological restoration, P−IBI showed a positive ecological response with the restoration of flow, showing an overall upward trend. However, with the flow regime increasing, P−IBI showed a hump-shaped downward trend. Based on this, the ecological response relationship between P−IBI and river flow in a statistical sense was constructed: y = (5.94 ± 0.33) × x + (−3743.35 ± 335.40) × x^2^ (R^2^ = 0.84). The 95% quantile of all P−IBI scores in the watershed was selected as the standard of watershed health. The environmental flow-threshold range of LID was 273–827 m^3^/s. This has an important reference significance for regulating environmental flow and improving the health and operation of the river ecosystem.

## Figures and Tables

**Figure 1 ijerph-20-00169-f001:**
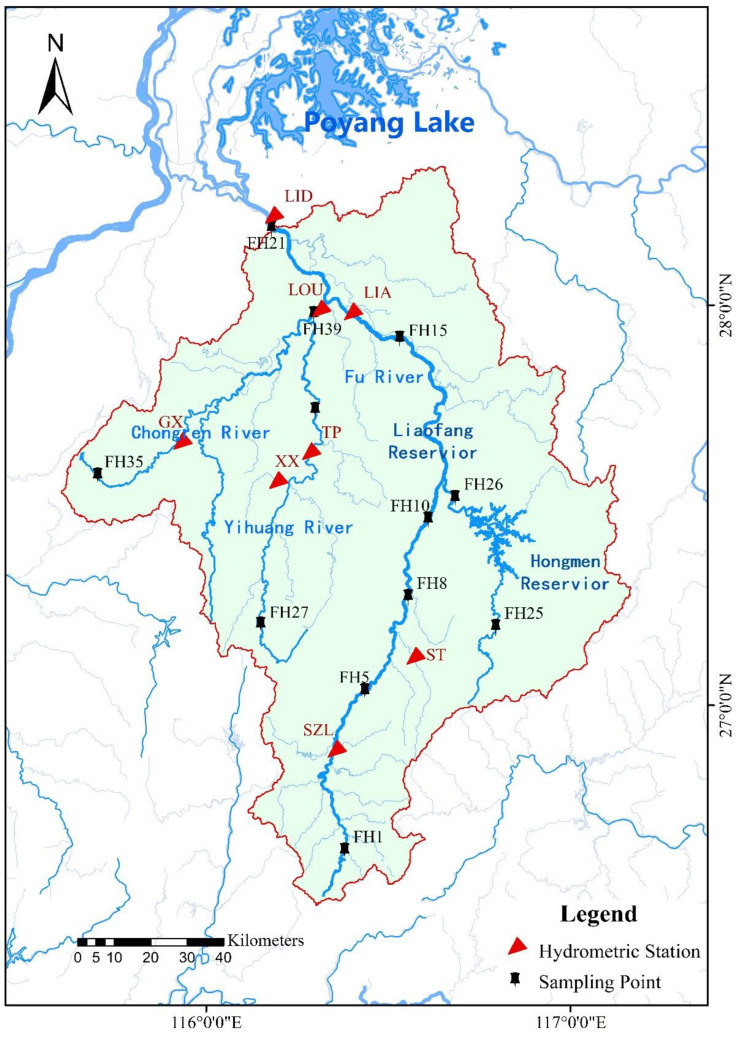
Distribution of hydrological stations in the Fuhe river basin.

**Figure 2 ijerph-20-00169-f002:**
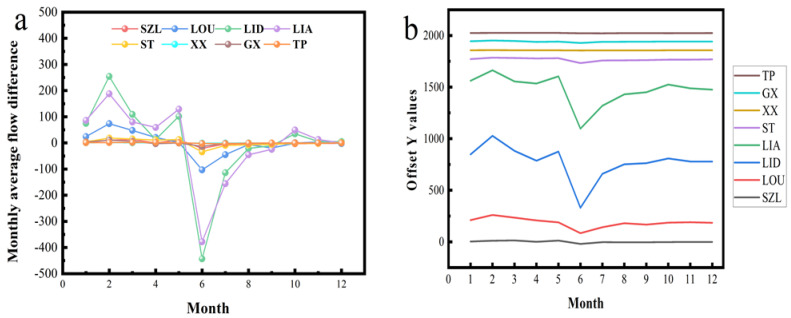
Monthly mean-flow deviation and Y−value deviation of each hydrological station. ((**a**): Monthy average flow difference; (**b**): Offset Y−value deviation).

**Figure 3 ijerph-20-00169-f003:**
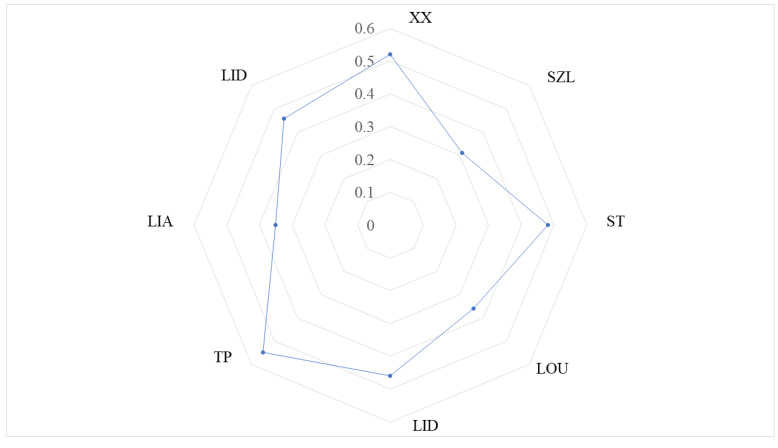
The overall alteration of hydrology before and after sluice construction in the typical control section of the mainstem.

**Figure 4 ijerph-20-00169-f004:**
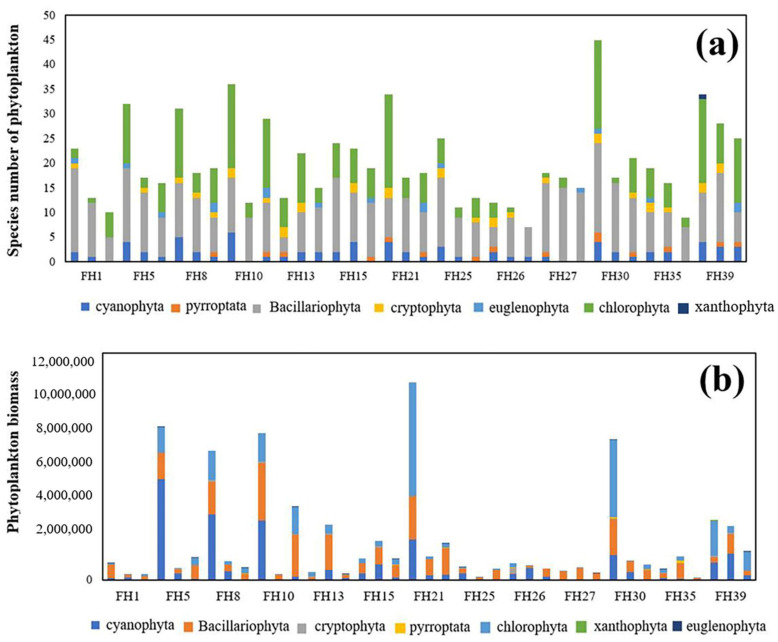
(**a**) Temporal and spatial distribution of phytoplankton species in Fuhe River Basin. (**b**) Spatial and temporal distribution of phytoplankton biomass in Fuhe River Basin (from left to right, each point is in order of high-water period, low-water period, and flat-water period).

**Figure 5 ijerph-20-00169-f005:**
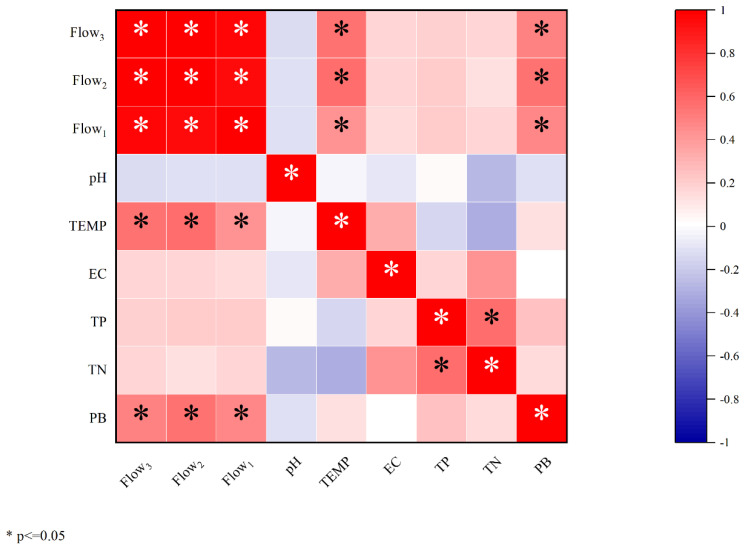
Spearman correlation heat map of environmental indicators and phytoplankton (* means the significant correlation between the two indicators, *p* < 0.05).

**Figure 6 ijerph-20-00169-f006:**
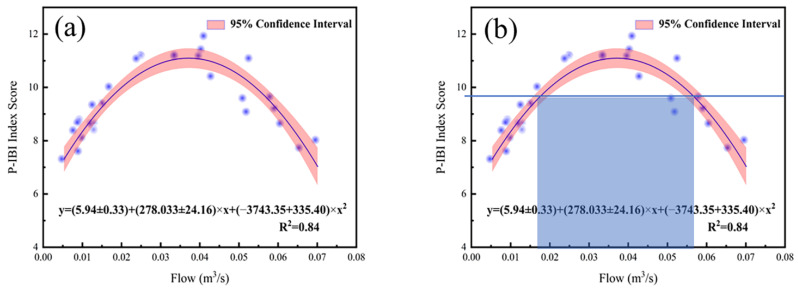
(**a**) Relationship between P−IBI index and flow response in the Fuhe River Basin; (**b**) flow–threshold range of downstream healthy ecosystems.

**Table 1 ijerph-20-00169-t001:** Candidate phytoplankton indicators of P−IBI in the Fuhe River Basin and their responses to disturbance.

Number	Biometrics	Response to Disturbance
M1	Number of algal species (total taxa)	Decrease
M2	Number of diatom species	Decrease
M3	Number of Dinophyta + Cryptophyta species	Increase
M4	Number of diatom species	Decrease
M5	Number of Dinophyta + Cryptophyta species	Increase
M6	Number of algal cells	Increase
M7	Dominant algal-cell density	Increase
M8	Bloom algae algal-cell density	Increase
M9	Dinophyta + Cryptophyta cell density	Increase
M10	Diatom algae-cell density	Decrease
M11	Dinophyta + Cryptophyta cell density	Increase
M12	Total biomass	Increase
M13	Diatom biomass	Increase
M14	Dinophyta + Cryptophyta biomass	Increase
M15	Cyanobacteria + Chlorophyta biomass	Increase
M16	Diatom biomass	Decrease
M17	Cyanobacteria + Chlorophyta biomass	Increase
M18	Dinophyta + Cryptophyta biomass	Increase
M19	The cell density of the TOP 3 dominant species	Increase
M20	Dominant algal-cell density	Increase
M21	Number of bloom-algae species	Increase
M22	Bloom-algae algal-cell density	Increase
M23	Shannon’s diversity index	Decrease
M24	Margalef index	Decrease
M25	Simpson index	Decrease
M26	Pielou index	Decrease

**Table 2 ijerph-20-00169-t002:** Index of hydrological alteration in the mainstem hydrological station of the Fuhe River. (**Note:** ↑ means the alteration is increasing; ↓ means the alteration is decreasing).

IHA	SZL	LOU	LID	LIA
	Alteration Rate and Situation	Alteration Rate and Situation	Alteration Rate and Situation	Alteration Rate and Situation
January average flow	14% (L) ↑	2% (L) ↑	49% (M) ↓	14% (L) ↑
February average flow	46% (M) ↑	53% (M) ↑	66% (M) ↓	45% (M) ↑
March average flow	29% (L) ↑	6% (L) ↑	15% (L) ↑	29% (L) ↑
April average flow	29% (L) ↑	73% (H) ↓	62% (M) ↓	18% (L) ↓
May average flow	4% (L) ↓	25% (L) ↓	28% (L) ↓	6% (L) ↓
June average flow	36% (M) ↓	18% (L) ↑	20% (L) ↑	10% (L) ↑
July average flow	46% (M) ↑	6% (L) ↑	23% (L) ↑	37% (M) ↑
August average flow	46% (M) ↑	37% (M) ↑	20% (L) ↓	10% (L) ↑
September average flow	4% (L) ↑	22% (L) ↑	88% (H) ↑	6% (L) ↑
October average flow	36% (M) ↑	49% (M) ↑	23% (L) ↓	22% (L) ↓
November average flow	46% (M) ↑	41% (M) ↑	32% (L) ↓	10% (L) ↓
December average flow	25% (L) ↑	29% (L) ↑	6% (L) ↑	2% (L) ↑
1-day minimum flow	4% (L) ↑	2% (L) ↑	37% (M) ↑	37% (M) ↑
3-day minimum flow	18% (L) ↓	14% (L) ↑	71% (H) ↓	53% (M) ↑
7-day minimum flow	7% (L) ↑	22% (L) ↑	105% (H) ↓	61% (M) ↑
30-day minimum flow	4% (L) ↑	61% (M) ↑	71% (H) ↓	61% (M) ↓
90-day minimum flow	39% (M) ↑	41% (M)	2% (L) ↓	14% (L) ↓
1-day maximum flow	25% (L) ↓	18% (L) ↓	20% (L) ↓	37% (M) ↓
3-day maximum flow	4% (L) ↓	33% (L) ↓	28% (L) ↓	53% (M)
7-day maximum flow	14% (L) ↓	33% (L) ↓	45% (M) ↓	29% (L) ↓
30-day maximum flow	7% (L) ↓	25% (L) ↓	28% (L) ↓	2% (L) ↓
90-day maximum flow	46% (M) ↓	18% (L) ↓	28% (L) ↓	22% (L) ↓
Number of zero-flow days	0% (L) ↓	0% (L) ↓	0% (L) ↓	0% (L) ↓
Base flow	25% (L) ↑	2% (L) ↑	88% (H) ↓	61% (M) ↑
Date of minimal flow	25% (L) ↓	14% (L) ↓	74% (H) ↑	45% (M) ↑
Date of maximal flow	33% (L) ↑	19% (L) ↓	11% (L) ↑	16% (L) ↑
Low-pulse flow count	25% (L) ↓	67% (H) ↓	22% (L) ↑	50% (M) ↑
Low-pulse flow duration	23% (L) ↓	61% (M) ↓	6% (L) ↑	12% (L) ↓
High-pulse flow count	5% (L) ↓	55% (M) ↑	16% (L) ↓	2% (L) ↑
High-pulse flow duration	27% (L) ↑	10% (L) ↑	45% (M) ↑	60% (M) ↓
Rise rate	23% (L) ↓	29% (L) ↓	11% (L) ↑	42% (M) ↑
Fall rate	68% (H) ↑	10% (L) ↓	37% (M) ↓	37% (M) ↓
Number of reversals	52% (M) ↑	53% (M) ↑	32% (L) ↑	6% (L) ↑

**Table 3 ijerph-20-00169-t003:** Index of hydrological alteration in the hydrological stations of the tributaries of the Fuhe River. (**Note:** ↑ means the alteration is increasing; ↓ means the alteration is decreasing).

IHA	ST	XX	GX	TP
	Alteration Rate and Situation	Alteration Rate and Situation	Alteration Rate and Situation	Alteration Rate and Situation
January average flow	8% (L) ↑	8% (L) ↓	28% (L) ↓	8% (L) ↑
February average flow	33% (L) ↑	8% (L) ↓	4% (L) ↑	16% (L) ↓
March average flow	50% (M) ↑	72% (H) ↓	44% (M) ↑	128% (H) ↓
April average flow	83% (H) ↓	8% (L) ↓	76% (H) ↓	56% (M) ↓
May average flow	25% (L) ↑	72% (H) ↓	32% (L) ↓	68% (H) ↓
June average flow	83% (H) ↓	72% (H) ↑	44% (M) ↑	20% (L) ↑
July average flow	33% (L) ↑	8% (L) ↑	20% (L) ↓	32% (L) ↑
August average flow	25% (L) ↑	66% (M) ↑	52% (M) ↑	64% (M) ↑
September average flow	50% (M) ↑	3% (L) ↑	8% (L) ↑	44% (M) ↑
October average flow	58% (M) ↓	43% (M) ↑	92% (H) ↑	8% (L) ↑
November average flow	58% (M) ↓	66% (M) ↑	32% (L) ↑	20% (L) ↑
December average flow	67% (H) ↑	3% (L) ↑	20% (L) ↑	64% (M) ↑
1-day minimum flow	25% (L) ↑	49% (M) ↑	28% (L) ↑	100% (H) ↓
3-day minimum flow	50% (M) ↑	83% (H) ↑	28% (L) ↑	64% (M) ↑
7-day minimum flow	8% (L) ↑	60% (M) ↑	40% (M) ↑	64% (M) ↑
30-day minimum flow	67% (H) ↑	26% (L) ↑	16% (L) ↑	52% (M) ↑
90-day minimum flow	50% (M) ↑	8% (L) ↑	80% (H) ↑	8% (L) ↑
1-day maximum flow	58% (M) ↑	43 (M) ↓	56%(M) ↑	7% (L) ↑
3-day maximum flow	67% (H) ↑	26% (L) ↓	32% (L) ↑	4% (L) ↑
7-day maximum flow	8% (L) ↑	72% (H)↓	80% (H) ↑	8% (L) ↑
30-day maximum flow	50% (M) ↑	49% (M) ↑	56% (M) ↑	32% (L) ↓
90-day maximum flow	8% (L) ↑	83% (H) ↑	44% (M) ↑	68% (H) ↓
Number of zero-flow days	0% (L) ↓	0% (L) ↓	0% (L) ↓	0% (L) ↓
Base flow	42% (M) ↑	31% (L) ↓	76% (H) ↑	76% (H) ↑
Date of minimal flow	67% (H) ↓	100% (H) ↑	20% (L) ↓	20% (L) ↑
Date of maximal flow	17% (L) ↓	49% (M) ↓	68% (H) ↑	104% (H) ↑
Low-pulse flow count	22% (L) ↓	59% (M) ↓	23% (L) ↓	62% (M) ↓
Low-pulse flow duration	63% (M) ↓	59% (M) ↑	25% (L) ↑	71% (H) ↓
High-pulse flow count	42% (M) ↑	20% (L) ↑	44% (M) ↓	15% (L) ↑
High-pulse flow duration	3% (L) ↑	48% (M) ↑	4% (L) ↑	4% (L) ↓
Rise rate	33% (L) ↓	49% (M)↑	4% (L) ↓	4% (L) ↓
Fall rate	17% (L) ↑	8% (L) ↑	56% (M) ↑	7% (L) ↓
Number of reversals	75% (H) ↑	69% (H) ↓	4% (L) ↑	79% (H) ↑

## Data Availability

The dataset analyzed during this study is available from the corresponding author upon reasonable request.

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
