# Peer review of "Determining Critical Thresholds of Environmental Flow Restoration Based on Planktonic Index of Biotic Integrity (P−IBI): A Case Study in the Typical Tributaries of Poyang Lake"

_ijerph, 2022, doi:10.3390/ijerph20010169_

Round 1

Reviewer 1 Report

The text is interesting from both the scientific and the factual aspects.  A few moderate errors are in the figures. I recommend publishing after correcting these errors.

The quantitative response relationship between the water area flow is described by a polynomial. Please write its formula correctly. Please write R2 to two decimal places. Deviations from the trend line should be expressed as confidence intervals. Add a description of the confidence intervals to the methodology.

Please write the curve in Figures 5 and 6 correctly.

Author Response

Dear Reviewer #1
Thank you for your letter and comments on our manuscript titled [Determining critical thresholds of environmental flow restoration based on Planktonic Index of Biotic Integrity (P-IBI): A case study in the typical tributaries of Poyang Lake] (Manuscript ID: ijerph-2073505). These comments helped us improve our manuscript, and provided important guidance for future research.

We have addressed the Reviewer’s comments to the best of our abilities, and revised text to meet the [International Journal of Environmental Research and Public Health] style requirements. We hope this meets your requirements for a publication.

We used a redaction pattern and marked the redaction in the manuscript in red. The main comments and our specific responses are detailed below:

Response to the reviewer`s comments:

The text is interesting from both the scientific and the factual aspects.  A few moderate errors are in the figures. I recommend publishing after correcting these errors.

Response: Thank you very much for your recognition of our article, we will try our best to modify to meet the requirements of the journal.

1. The quantitative response relationship between the water area flow is described by a polynomial. Please write its formula correctly. Please write R2 to two decimal places(L361-362).

Response: Thank you very much for your advice. We have corrected the formula and left R2 with two decimal places.

2. Deviations from the trend line should be expressed as confidence intervals. Add a description of the confidence intervals to the methodology.

Response: Thank you very much for your valuable comments. We have added the confidence interval (Fig.5). And added a description to the methods section (L172-173).

3. Please write the curve in Figures 5 and 6 correctly.

Response:  Thank you very much for your careful question. We have corrected the formulas in Figure 5 and Figure 6 and combined them into one figure (Fig.5).

Reviewer 2 Report

Based on the analysis of the relationship between river phytoplankton integrity index and runoff, taking Fuhe River Basin as an example, based on the analysis of the hydrological situation of the main stream and tributaries, this paper reveals that the flow is the main factor affecting phytoplankton, especially the flow of two months has the highest correlation, and establishes a tool for rapid assessment of river ecosystem health based on the relationship between phytoplankton biological integrity index and changes in river environmental factors. The range of environmental flow threshold for river health is determined, which has important reference significance for regulating river environmental flow and improving river ecosystem health.

1. There are 9 hydrological stations mentioned in the regional overview, but only 8 stations are marked in Figure 1, and there is no FT station. Subsequent analysis and evaluation are also based on 8 stations, which need to be checked.

2. The specific description of phytoplankton survey is not very detailed and should be supplemented and improved.

3.Figures 5 and 6 can be combined into one picture.

Author Response

Dear Reviewer #2
Thank you for your letter and comments on our manuscript titled [Determining critical thresholds of environmental flow restoration based on Planktonic Index of Biotic Integrity (P-IBI): A case study in the typical tributaries of Poyang Lake] (Manuscript ID: ijerph-2073505). These comments helped us improve our manuscript, and provided important guidance for future research.

We have addressed the reviewer’s comments to the best of our abilities, and revised text to meet the [International Journal of Environmental Research and Public Health] style requirements. We hope this meets your requirements for a publication.

We used a redaction pattern and marked the redaction in the manuscript in red. The main comments and our specific responses are detailed below:

Response to the reviewer`s comments:

1. There are 9 hydrological stations mentioned in the regional overview, but only 8 stations are marked in Figure 1, and there is no FT station. Subsequent analysis and evaluation are also based on 8 stations, which need to be checked.

Response: Thank you very much for your careful review. I'm very sorry that this is our mistake. We have revised the number of hydrographic stations from 9 to 8 (19). And delete the FT station (L97).

2. The specific description of phytoplankton survey is not very detailed and should be supplemented and improved.

Response: Thank you very much for your valuable comments, we have added the description and details of the phytoplankton survey (L152-162).

3. Figures 5 and 6 can be combined into one picture.

Response: Thank you very much for your advice, we have merged Figures 5 and 6 (Fig.5).